# Relationship among Self-Efficacy Expectations, Locus of Control, and Attributions in Bariatric Patients

**DOI:** 10.3390/ijerph19127107

**Published:** 2022-06-09

**Authors:** Carla Ugarte, Álvaro Quiñones, Luis Angel Saúl

**Affiliations:** 1Department of Social Science, University of Tarapacá, Iquique 1101783, Chile; cugartep@gmail.com (C.U.); alvaroquinonesb@gmail.com (Á.Q.); 2Faculty of Psychology, Universidad Nacional de Educación a Distancia (UNED), 28040 Madrid, Spain

**Keywords:** sleeve gastrectomy, gastric bypass, weight regain, mixed design

## Abstract

Background: Bariatric surgery is the most effective method for achieving accelerated weight loss. However, in the short- and medium-term, between 20% and 40% of patients regain a significant percentage of the weight lost. Cognitive and attitudinal psychological variables contribute to explaining weight regain. The aim of this study was to analyze differences in self-efficacy, locus of control, and attributions among bariatric patients, in accordance with weight maintenance or weight regain. Methods: Participants were classified according to weight regain (≥15% weight regain) and weight maintenance (<15% weight regain). A receiver operating characteristic (ROC) curve analysis was employed to assess the diagnostic value of the locus of control for weight loss and to establish a cutoff point to differentiate those who maintained weight loss from those who regained more than 15% of the weight lost. Results: Those who maintained weight loss showed a statistically higher locus of control ratio than those who regained weight. The locus of control ratio was associated with a lower risk of weight regain (odds ratio 0.760, *p* = 0.018). Using the area under the ROC curve (AUC), the locus of control significantly identified those who maintained weight (AUC = 0.761; *p* = 0.001). The maximum combination of sensitivity and specificity was shown at the cutoff point of 39. Qualitative results show a difference in the type of attributions and expectations according to current weight maintenance or weight regain status. Conclusion: Participants’ self-efficacy expectations, locus of control, and attributions change in accordance with the outcome achieved in terms of weight regain or weight maintenance.

## 1. Introduction

Bariatric surgery is the most effective treatment for obesity [1,2]. However, since obesity is a chronic disease, unless it is accompanied by long-term lifestyle changes, surgery alone often results in relapse [3,4]. Between 20% and 40% of patients do not lose the expected percentage of excess weight or regain a significant part of it in the medium-term [5,6]. In terms of the variables that influence these sub-optimal outcomes, research has shown that psychological, behavioral, and social factors [7,8,9,10] are those that contribute most to explaining weight regain, defined as gaining 15% or more of the weight lost from nadir [11,12].

For some years now, various authors have been studying certain psychological variables of a cognitive-attitudinal nature, such as self-efficacy and locus of control, as well as, to a lesser extent, attributions and their association with body weight regulation, specifically in terms of adherence to the lifestyle changes required to maintain weight loss [13,14,15].

“Self-efficacy” [16,17] is defined as the judgments people make about their own ability to achieve a desired outcome. Self-efficacy comprises two types of expectations: efficacy and outcome. The first refers to one’s assessment of one’s own ability to successfully carry out an action, and the second to one’s belief that engaging in said behavior will enable one to obtain the desired outcome. Perceived self-efficacy for weight loss refers to one’s beliefs about one’s ability to organize and carry through with the courses of action required to control one’s body weight [18].

Self-efficacy has been found to predict success in both losing and maintaining body weight [19,20,21,22], as well as the intention to engage in healthy eating behaviors and physical exercise [23]. It has also been positively associated with weight loss at the start of treatment [24] and greater food self-efficacy has been linked to profound weight loss among adults after surgery [25].

“Locus of control” (LC) was first described by Rotter [26] and refers to one’s personal beliefs regarding the degree of control one has over life events. The construct distinguishes between two types of locus: internal (ILC) and external (ELC). An ILC reflects the belief that events are contingent on one’s own behavior, whereas an ELC reflects the belief that events are not contingent on one’s own actions, but rather dependent on luck, coincidence, fate, or the power of others. The conceptualization of LC suggests that behavior depends on how much control each person perceives themselves as having, and that, in turn, their perception of control determines the possibility of predicting and controlling human behavior. Consequently, those with an ILC assume more responsibility in relation to life events, and this in turn is linked to the idea that LC may affect chronic diseases, such as obesity [14,27].

More specifically, existing evidence suggests a positive association between having an ILC and successfully completing a weight-loss program and maintaining a 10% reduction in body weight [14,15,28]. An ILC has also been associated with increased motivation to lose weight among overweight/obese individuals after reading educational health material and engaging in physical activity [14], something which was not observed among individuals with an ELC [29]. Among bariatric patients, previous studies have found differences in the association between LC and weight loss [21,30].

Weiner’s attribution theory [31] defines attribution as the causal explanation of an important life event or outcome. In the context of bariatric patients, it is defined as the causal explanation of the outcomes obtained before, during, and after surgery. Attribution is made up of three dimensions: (a) LC of the outcome causes (internal: the outcome is caused by factors inherent to the individual; external: the outcome depends on environmental factors); (b) stability of the causes (stable: causes have a consistent influence; unstable: causes are inconsistent and only have an influence on occasions); (c) degree of perceived control over the outcome causes (controllable: causes are perceived as modifiable by an act of will; uncontrollable: causes are perceived as unmodifiable).

Qualitative studies in this field have provided valuable information about the course of obesity, as well as the motivations and variables associated with weight maintenance and regain among bariatric patients. It is well-known that patients who undergo surgery have often experienced weight-related problems since childhood and/or adolescence [32,33]. Most have a history of unsuccessful attempts to lose weight using other methods [32,34], and the principal reasons prompting them to opt for surgery are linked to health, quality of life, and wellbeing [35,36].

The present study offers quantitative and qualitative data to further our understanding of weight maintenance and regain among bariatric patients, combining information gathered from psychometric instruments with an analysis of narrated experiences and the explanations provided by participants regarding their outcomes.

## 2. Method

### 2.1. Participants

For the quantitative analysis, we used a non-probabilistic convenience sampling method to recruit 97 patients who had undergone bariatric surgery between 2002 and 2017. For the qualitative analysis, participants were invited to participate following a convenience sampling. Thirteen people agreed to be interviewed and it was considered that a percentage close to 20% of the total sample was sufficient for this analysis, as the results of the qualitative analysis allowed us to identify and analyze the explanations patients have for their current state of weight maintenance or weight regain. The inclusion criteria were: men and women aged between 18 and 65 years who had undergone bariatric surgery (gastric bypass or sleeve gastrectomy). Exclusion criteria were: having received surgical treatment other than a gastric bypass or sleeve gastrectomy, having undergone a secondary procedure (revision surgery), and in the case of women, having recently been pregnant or breastfeeding.

### 2.2. Instruments

A sociodemographic and biomedical questionnaire was designed ad hoc for the present study. Data were collected regarding participants’ age, sex, marital status, and education level, among others. The biomedical variables included were: height, previous weight, previous nadir, current weight, type of surgery, and year of surgery, among others. This information was obtained directly from the participants. There was no access to patients’ medical records.

Self-efficacy for losing weight [37]: Responses were given on a 7-point Likert-type scale ranging from 0 (I cannot do it) to 6 (I am sure I can do it). The scale measures three dimensions of self-efficacy: self-efficacy for dieting (15 items), self-efficacy for exercising (13 items), and self-efficacy for buying diet-friendly food (10 items). The scale had good construct validity and high levels of internal consistency (0.93 to 0.94).

Locus of control regarding eating behavior and weight gain [38]: The scale measures respondents’ perceptions of control in relation to eating, gaining weight, and losing weight, and has a 4-point Likert-type response scale ranging from 1 (totally disagree) to 4 (totally agree). Each dimension measures both ILC and ELC. Items measuring ELC are reverse coded, with higher scores indicating a stronger ILC. The scale has good construct validity and high levels of internal consistency (0.85 to 0.93).

Current weight loss status: operationalized as a dichotomous variable: maintaining or regaining. To calculate the percentage of regain, a simple delta was created of the difference between TWL% at nadir and TWL% at the time of the study (TWL% formula: [(initial weight − current weight)/(initial weight)] × 100) [39]. “Maintaining” was defined as the regaining of less than 15% of the total weight loss (TWL) percentage from nadir. “Regaining” was defined as the regaining of 15% or more of the TWL% from nadir [11,12].

Qualitative data were gathered by means of semi-structured interviews designed specifically for this study [40]. Prior to each interview, the interviewer prepared a thematic script with open-ended questions to guide the interviewee. In accordance with the aim of the study, three areas of analysis were covered: “weight history prior to the surgery”, “motivation/expectations regarding the bariatric process”, and “current status” (maintaining or regaining) (Table 1).

### 2.3. Procedure

The study was approved by the Ethics Committee at the University of Concepción and the Scientific Ethics Committee of the Concepción Health Service.

Quantitative data were collected between April 2016 and August 2018. Participants were recruited from monitoring sessions with the Obesity Treatment Team (ETO-Conce) and diverse other teams in Santiago. Participation was voluntary and informed consent was obtained from all participants. All instruments were self-administered.

For the qualitative analysis, participants were contacted over the phone. Interviews were held in a place in which participants felt comfortable (their home or place of work) between August and December 2018. Only the interviewer and the participant were present during the interviews. All interviews were carried out in accordance with the interview protocol (Table 1) and lasted for an average of 66 min (a minimum of 54 min and a maximum of 87 min).

Participants’ responses were audio-recorded, transcribed, and analyzed using ATLAS.ti (V.6.2). The coding process was carried out independently by two members of the research team, and the data analysis was deemed to have concluded when all researchers agreed that the saturation point had been reached.

### 2.4. Data Analysis

To compare groups in accordance with current status (regaining versus maintaining), a one-way MANOVA was performed to analyze the various dependent measures simultaneously [41]. Logistic regression analysis was applied to evaluate the potential predictive ability of psychological variables in maintaining weight loss.

The discriminative ability of the locus of control ratio for weight loss was estimated between those who maintained weight loss and those who regained more than 15% of the weight lost from nadir with the area under the receiver operating characteristic curve (AUC) [42].

For the qualitative analysis, a thematic content analysis was carried out with pre-established (top-down) categories (codes). It is worth highlighting that the aim of “content analysis” techniques is to identify and explain the cognitive representations that make sense of the entire communicative narrative [43]. Top-down coding was based on existing literature, clinical experience, personal knowledge of people undergoing bariatric surgery, and the objective of the qualitative component “to understand through the identification of attributions and expectations the bariatric process and its outcomes”. On this basis, the top-down coding method was considered the most appropriate to identify and analyze whether patients’ narrative data could be understood within the framework of Weiner’s attribution theory [31], Bandura’s self-efficacy theory [16], and stages of change [44]. Thus, coding gave rise to two types of codes: first-order and second-order. For first-order coding, a total of 10 categories were generated on the basis of Weiner’s attribution theory [31] (1985) and Bandura’s self-efficacy theory [16] (1977) and were applied to the interviews in accordance with each specific area of analysis. Subsequently, for the second-order coding, a total of seven codes were generated and applied only to the third area of analysis, in accordance with the various stages of change [44] (see Table 2).

The unit of analysis for both coding processes was the paragraph.

Before coding the 13 interviews, a hermeneutic unit was created with the information. Next, the codes were generated using the Atlas.Ti software program (V.6.2). The characteristics of the codes are presented in Figure 1.

The analysis was then divided into two phases. First, the (top-down) first-order coding process was performed independently on participants’ responses, applying only one category to each answer (see example in Figure 2). This coding system had an inter-observer agreement value within the range generally considered as good (*K* = 0.77 to *K =* 0.85) [45]. The independent reviewers discussed each area of disagreement until a consensus was reached.

Translation: [1:7] (186) *“I think I’d like to have another operation, so they can tighten things up a bit more…so I can get that feeling of being full that I used to get but don’t any more. It’s been two years since the operation…and for a year, more or less, I would get that feeling, you know, of being stuffed all of a sudden. I was worried because I thought they hadn’t done the operation properly”. External, stable and uncontrollable attribution* (c2 N3).

Next, the second-order coding process was applied solely to the third area of analysis. For the second-order coding, the inter-observer agreement value was between *K* = 0.73 and *K* = 0.82 [43]. Each area of disagreement was then discussed until consensus was reached.

Credibility, transferability, reliability, and confirmability criteria were all taken into consideration [46]. To improve credibility, the interpretations of the interviews were checked and confirmed. Transferability was ensured through detailed descriptions of the participants’ experiences and contexts during the data collection process. Reliability was guaranteed by means of detailed descriptions of each methodological decision made during the research process. Finally, confirmability was ensured through the inclusion of detailed descriptions with extracts from the data collected.

## 3. Results

Table 3 offers descriptions of the participants in the study, in accordance with diverse sociodemographic and biomedical variables. It also specifies whether or not they were maintaining or regaining weight at the time of the study.

A multivariate analysis of variance (MANOVA) was performed to explore the differences between groups in the different dimensions of self-efficacy and LC, adjusting for sex, age, and years since surgery. Box’s M and Levene’s tests were performed to verify the assumptions. The results of the Box’s M test were not significant (Box’s M = 44.60, F = 1.324, *p* = 0.119). The results of the Levene’s test revealed homogeneity and equality of variance for the self-efficacy variables, sex and age (ps < 0.05), although not for the LC variables and years since surgery (ps > 0.05). These results indicate that the current weight status (maintaining versus regaining) had a significant multivariate effect on the dimensions of self-efficacy and LC [F(7.89) = 6.45, *p* < 0.001; Wilks’ Lambda = 0.663, partial η^2^ = 0.337] (see Table 4).

To further investigate the possible predictive ability of psychological variables and weight maintenance, a logistic regression analysis was performed. The likelihood ratio test (X^2^ = 36.971; gl = 7; *p* = 0.001) and the Hosmer and Lemeshow test (X^2^ = 2.284; gl = 8; *p* = 0.971) indicate that the model predictions correctly describe the data presented. Thus, after adjusting for age, sex, and years since surgery, the locus of control for losing weight appeared as a potential modifiable predictor (odds ratio 0.76 (95% CI: 0.606–0.953); *p* = 0.018). The model explained 52.4% of the variance found in maintaining weight loss (R^2^ = 0.524) (see Table 5).

The score obtained on this variable was used to assess the diagnostic power of the locus of control for losing weight to discriminate between patients who maintained the loss from those who regained 15% or more of the weight from nadir. For this analysis, we only used the locus of control for the losing value since it is the only significant modifiable variable in the logistic regression model. The area under the ROC curve (AUC) of the LC was 0.761 (95% CI 0.63–0.88; *p* = 0.040) (Figure 3). According to the combination of maximum sensitivity and specificity calculated with the ROC curve, the optimal cut-off value for LC was 39 points. Using this optimal cut-off value for LC, the positive predictive value(s) (PPV) and negative predictive value(s) (NPV) were approximately 87% and 35%, respectively. To further validate the usefulness of the LC ratio as a new marker of weight loss maintenance, the patients included in the initial sample of the current study were reclassified. In this analysis, 75.3% of the patients were correctly classified (efficiency) by the locus of control ratio.

The receiver operating characteristic (ROC) curve for the locus of control for losing weight was used to discriminate between participants who maintained 15% or more weight loss since nadir from those who regained weight. The arrows indicate the location of the selected cut-off point with the best sensitivity and specificity. The data of sensitivity (S), specificity (Sp), positive predictive value (PPV), negative predictive value (NPV), and the proportion of patients correctly classified (efficiency) are shown.

### 3.1. Qualitative Results

The 13 participants who were recruited for the qualitative analysis had a mean age of 45.5 years (SD = 9.8). Their age at the time they underwent surgery was 40.3 years (SD = 10.9). Their BMI prior to the surgery was 45.7 kg/m^2^ (SD = 8.5 kg/m^2^) and at the time of the study it was 30.6 kg/m^2^ (SD = 7.3 kg/m^2^). One-third of the participants in the study (33.3%) reported weight regain.

The three topics and six subtopics covered resulted in descriptions of the surgical process and its outcomes and enabled an understanding of it through the identification of attributions and expectations (Table 6).

The characteristics of the interviewees are presented in Table 7.

#### 3.1.1. Weight History Prior to the Surgery

##### Explanations for Being Overweight

Participants described how, prior to the surgery, their bodies had gradually increased in size, in some cases since childhood, and in others since adolescence. Regardless of the time of onset of their weight issues, all participants attributed their previous excess weight to internal, stable causes perceived as uncontrollable at the time. Specifically, they attributed their increasing weight to emotional (anxiety, stress, distress) and, to a lesser extent, genetic causes. One participant expressed these attributions as follows:

*“To be honest, prior to the surgery it had been a life-long struggle [with obesity] ... linked to psychological problems, mood issues. Now I realize that food was for me like drugs are for many addicts—a means of escape, a way—a bad way—of coping with distress”*.(9NNR)

##### Explanations Regarding the Outcomes of Other Weight-Loss Methods

All participants claimed to have tried a wide range of strategies to slow their weight gain and even lose weight prior to the surgery. None of these strategies had outcomes that were sustained in the short- and/or long-term. The explanations given in relation to previous strategies and their outcomes were also linked to internal, stable, and uncontrollable causes. In this case, the principal causes were also emotional in nature, with participants highlighting the boredom associated with pre-established nutritional guidelines or non-pre-established diets. In the words of two participants:

*“I’d be fine for one or even maybe two months, but then I’d fall back again [into eating] because I got bored of always eating the same things. It’s impossible to always eat chicken and lettuce. So I’d quit (the diet) and would go back to eating like before, or perhaps even more, and my weight would go up again”*.(5NNR)

*“You just end up getting bored. I’d lose a couple of kilos while I was on the diet, or at first when I did acupuncture, but then it was really hard, almost impossible, to keep going. So I’d gain the weight back, and then some, and it was exhausting. At some point I just got so bored I said “hey, what does it matter?” It was very frustrating”*.(8NRE)

#### 3.1.2. Motivation/Expectations Regarding the Bariatric Process

##### Motivation/Expectations at the Start of the Bariatric Process

All participants saw the surgery as a means of achieving diverse results, such as health improvement, disease remission, and a better quality of life. Consequently, all had expectations regarding the outcome of the operation. In the words of two participants:

*“My main aim was to prolong my life and to not have diseases that were so complex. Because I’m really scared of diabetes and that was what prompted me to have the surgery”*.(2NRE)

*“I wanted to have a decent life, because before I found it really hard to walk ... everything was torture: climbing the stairs, doing the most mundane, everyday things that other people find easy ... it was all a major effort for me”*.(1NNR)

##### Evolution of Expectations Associated with the Surgery

Among participants who had managed to maintain their weight loss, a change in initial expectations was observed. Once food intake restriction was relaxed, expectations associated with the outcomes of the surgery turned into expectations regarding its effectiveness; or in other words, into perceptions regarding their ability to follow through with a specific course of action in order to achieve a desired outcome. Participants accepted that the surgery had “done its work”, but after a time, the outcome no longer depended on the procedure, but rather on their own ability to maintain a lifestyle that was different from the one they had had before. Therefore, expectations shifted from what the surgery could give them in terms of outcomes, to what they were capable of doing to maintain said outcomes, something they considered to be as (or even more) necessary than the surgery itself. One participant explained it as follows:

*“There are a load of physical signs, and it’s like you can forget about it because your body just tells you ... [food intake restriction], but then, after a while, your stomach expands again and you have to take responsibility. So, of course, at first I lost 70 kilos, but then the curve started to reverse and since I wasn’t really in touch with what was happening, I gained 10 kilos without realizing. And that’s when you say no, I can’t trust exclusively to the effects of the surgery. And I went back to thinking that surgery is only a support for a much more complex process ... And today I feel, I feel much more empowered; I’m much prouder of where I am now than of where I was straight after the surgery. Before, it was the result of the surgery, but now, to a large extent, it’s down to the things I’ve done rather than to just the staples I have in my belly”*.(4NNR)

This shift in expectations was not observed among those who had regained 15% or more of their TWL%. Among this group, outcome expectations continued to be prevalent, and these people remained focused on food intake restriction, with some even considering a second bariatric procedure. According to one participant:

*“The weight loss I achieved with the surgery makes me want to have another one ... I think it’s the best solution”*.(13NRE)

#### 3.1.3. Current Status

##### First-Order Coding—Explanations Regarding Current Status

Differences were observed in the explanations given by participants regarding their current status (maintaining/regaining). Participants who had regained 15% or more of their TWL% generally attributed the outcome of the surgery at the moment of the study to external, stable, and uncontrollable causes. These explanations were linked to the surgery itself, lack of perceived support for maintaining a healthy lifestyle, and other external problems, such as personal finances, which prevented them from engaging in “healthy” eating habits. In the words of two participants:

*“What happened was that food intake restriction didn’t last long and I thought the surgery hadn’t worked. I remember that a few months after the procedure, I asked the nutritionist what would happen if I ate two boiled eggs instead of one. She told me it would be impossible, that even if I wanted to eat two, my stomach wouldn’t be able to take it. And I replied, well it can—it can even take three!”*.(2NRE)

*“The truth is I lacked support. At first I had a lot [of support] but then it was just me and my new habits against everyone [family] and in the end you give in. If I had had the support of my family, I’m sure things would have been different”*.(7NRE)

In contrast, participants who had managed to maintain their weight loss made internal, stable, and controllable attributions, emphasizing that while the surgery had helped them, their ability to maintain the outcome currently depended on their behavior, attention, and concern regarding their diets, as well as on the constant monitoring of their status. These participants were constantly aware that they needed to look after themselves in order to avoid relapsing to their previous state. One participant expressed it in the following manner:

*“I think that, with things like this, you can’t rely on anyone else. The support of your family and friends is important, but at the end of the day you have to have the ability to self-regulate, because everything ultimately depends on self-regulation”*.(6NNR)

##### Second-Order Coding—Stages of Change

According to Prochaska and DiClemente’s stages of change [43], participants who were regaining were in either the contemplation or the preparation stages. For example, as one participant put it:

*“Look, I know perfectly well what eating healthily means; I know what I should and should not be eating, what is good for me and what will keep me healthy and help me lose weight. And at this moment in time, my (eating) patterns are clearly not healthy”*.(7NRE)

In contrast, participants who were maintaining their weight loss were mainly in the action and maintenance stages. For example, as one participant put it:

*“I’ve gradually been taking charge of this. I mean, I started going to a psychologist because I decided it was best; I found an expert gastroenterologist to help me control myself (medically/metabolically) and I sat down and decided to change my diet, and that’s what I’m working on now”*.(3NNR)

Participants who were maintaining, however, had not yet reached the consolidation stage of change. This was because they claimed that what enabled them to maintain the changes was the fact that they saw the whole enterprise as an ongoing process. In relation to this, one participant stated that:

*“This is a lifelong process. I’m fine now, I feel well and there may even be some days in which it’s not an effort, but others days it is. I know I have to remain alert to my habits, because it’s very easy to get distracted and start relapsing”*.(12NNR)

## 4. Discussion

Our research evaluated the relationship between weight maintenance and weight regain after bariatric surgery and the psychological variables—locus of control, self-efficacy, and attributions. Therefore, what follows should be understood with the limitations that imply not having evaluated maintenance–gain from a multidimensional approach, considering, for example, health determinants, genetics, homeostatic balance, and other factors that influence weight and the development of obesity.

Our results reveal an association between the outcomes of bariatric surgery in terms of current status (maintaining weight from nadir) and the LC, self-efficacy, and attribution variables. Both the quantitative and qualitative analyses returned similar results from different perspectives, integrating both the psychometric measures and the thematic analysis of participants’ narratives.

In general terms, around 20% of participants in the quantitative analysis and 33.3% (*n* = 4) of those in the qualitative analysis reported weight regain from nadir, a finding that is consistent with the current literature in this field [6,47].

First, the quantitative analyses regarding self-efficacy distinguished between participants who were maintaining their weight loss from nadir and those who were regaining it. This is consistent with that found also in the qualitative analysis, in which it was observed that those who were maintaining their weight loss had also shifted from outcome expectations to efficacy expectations. The same was not true for participants who had regained weight, who remained focused on expectations of the outcomes “promised by the surgery”, without accepting any internal agency in the process. These associations are important because they help identify variables that may have an impact on the post-bariatric process in terms of maintaining or regaining the weight lost. The quantitative and qualitative results found reveal the importance of bariatric patients developing a sense of self-efficacy, which may be a key focus of therapeutic work with this population, particularly when food intake restrictions start to become more relaxed and patients need to restructure their habits in order to integrate different food groups into their diets.

Second, the quantitative results indicate that LC also distinguished between participants who were maintaining and those who were significantly regaining, with this variable alone explaining 25.6% of the variance observed in the current status. Higher LC scores always imply an ILC, and participants who were maintaining differed from those who were regaining in terms of ILC and ELC. This is consistent with the qualitative results found, which revealed that participants who were maintaining their weight loss had shifted from internal, stable, and uncontrollable attributions in relation to their previous weight history and unsatisfactory results with other intervention models, to internal, stable, and controllable attributions regarding their current outcomes in terms of weight loss. This finding serves to highlight the importance of not only patients becoming involved in their own process, but also the way in which they do so. It is therefore vital to help bariatric patients gain a better understanding of the fact that they are the “protagonists and spectators of their own work” in terms of losing weight and then maintaining said weight loss. These results have implications for a multidisciplinary therapeutic approach. Specifically, psychotherapeutic work with bariatric patients should encompass a comprehensive analysis and should aim to help them generate and incorporate the ability to permanently self-monitor. This means helping them understand that obesity is a chronic disease and that any doubts they may have at any time are important and expected and should be shared with the multidisciplinary team so that the best help can be provided in each individual case. Furthermore, the members of the multidisciplinary team should seek to convey to patients at all times that in order to maintain the changes achieved, they must adopt an attitude of constant reflection in relation to their “weight problems”, and remain alert to any difficulties. This in turn will mainly depend on their commitment to themselves, their awareness of their weaknesses, and their capacity to regain self-control when they lose it. Similarly, it is also very important to help patients understand that surgery is only an aid in a complex process that has a past, present, and future in their lives, and is called “Struggling with weight in a positive, committed manner”. Consequently, successfully maintaining weight loss will depend on their ability to safeguard their new eating habits and to engage in the prescribed physical activities throughout the whole of their chosen “new life”, which “has not been imposed” on them and to which they are committed.

### Limitations

The results of the present study only apply to the sample and cannot be generalized to other groups or contexts. The cross-sectional and relational nature of the quantitative analyses precludes any causal relationship being established between the study variables, and the results obtained should therefore be considered with caution. Another limitation is the fact that the sample group mainly comprised women (89.7%).

## 5. Conclusions

The present study focuses on the personal characteristics of patients who have undergone bariatric surgery and their associations with outcomes in terms of maintaining the weight lost. We found that those with a higher internal locus of control, self-efficacy, and efficacy expectation levels reported better outcomes in terms of weight loss maintenance in the medium- and long-term. These findings highlight the importance of empowering people with obesity, since “being aware of their condition and their ability to maintain a viable level of health” helps them become the protagonists of the changes they need to make, rather than mere spectators of their chronic obesity.

It is important to highlight the suitability of the mixed research design used. The qualitative analyses enrich our comprehension of the results returned by the quantitative ones, and in our opinion, are highly relevant to gaining a more specific understanding of people who undergo bariatric surgery as a means of promoting their health.

## Figures and Tables

**Figure 1 ijerph-19-07107-f001:**
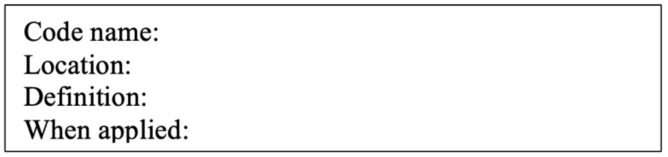
Code characteristics.

**Figure 2 ijerph-19-07107-f002:**
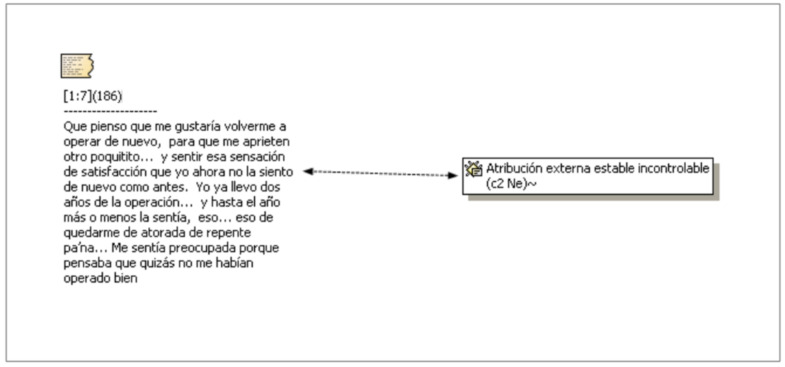
Example of coding with its corresponding excerpt.

**Figure 3 ijerph-19-07107-f003:**
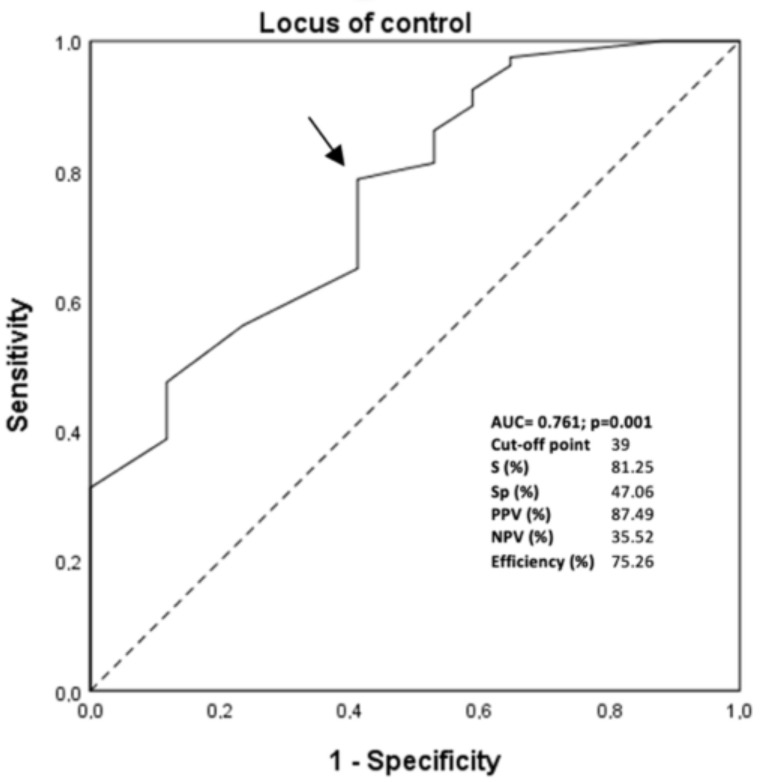
The ROC curve for the modifiable predictor of locus of control for losing weight.

**Table 1 ijerph-19-07107-t001:** Interview script and areas of analysis.

Area of Analysis	Interview Script
Weight history prior to the surgery	How would you describe your weight history prior to the surgery?
	To which factors do you attribute your weight problems at that time?
	Prior to the surgery, did you try any other methods of losing weight?
	Can you tell me about your previous attempts and outcomes?
Motivation/expectations regarding the bariatric process	What prompted you to think about having the surgery?
	Why did you decide to have the surgery, and what were your goals?
	What can you tell me about the expected outcome versus the outcome obtained?
Current status	How did the surgery go?
	After the first 6 months, what was your diet like?
	What factors do you think helped you maintain the weight loss achieved? *
	What factors do you think did not help you maintain the weight loss achieved? *
	To which factors would you attribute your weight regain?
	In relation to your process, what current challenges do you face?

Note: * question asked in accordance with the current weight loss status.

**Table 2 ijerph-19-07107-t002:** Thematic matrix applied in accordance with the area of analysis.

**First-Order Coding**		
**Area of Analysis**	**Variable**	**Code**
Weight history prior to the surgery	Attributions	Stable, internal, and controllable
		Stable, internal, and uncontrollable
		Unstable, internal, and controllable
		Unstable, internal, and uncontrollable
		Stable, external, and controllable
		Stable, external, and uncontrollable
		Unstable, external, and controllable
		Unstable, external, and uncontrollable
Motivation/expectations regarding the bariatric process	Self-efficacy	Efficacy expectations
		Outcome expectations
Current status	Attributions	Stable, internal, and controllable
		Stable, internal, and uncontrollable
		Unstable, internal, and controllable
		Unstable, internal, and uncontrollable
		Stable, external, and controllable
		Stable, external, and uncontrollable
		Unstable, external, and controllable
		Unstable, external, and uncontrollable
**Second-order coding**		
**Area of analysis**	**Variable**	**Code**
Current status	Stages of change	Pre-contemplation
		Contemplation
		Preparation
		Action
		Maintenance
		Consolidation
		Relapse

**Table 3 ijerph-19-07107-t003:** Sociodemographic variables in accordance with weight loss status (regaining or maintaining).

	Bariatric Patients			Statistics
	Total M (SD) *n* = 97	Regaining M (SD) *n* = 17	Maintaining M (SD) *n* = 80	
Age	39.78 (10.25)	35.53 (10.63)	40.6 (10)	*t =* 1.91; *p* = 0.059
Sex (female)	89.7% (87)	82.4% (14)	91.3% (73)	X^2^ *=* 1.2; *p = 0.27*
Previous medical conditions	M (SD)	M (SD)	M (SD)	
Pre-surgical BMI	41.17 (6.3)	41.44 (7.6)	41.11 (6.1)	*t* = −0.192; *p* = 0.85
Nadir BMI	24.7 (4.6)	24.34 (4.7)	24.79 (4.6)	*t* = 0.365; *p* = 0.72
Current BMI	28.0 (5.6)	33.12 (7.8)	26.93 (4.3)	*t* = −3.14; *p* = 0.006
Years since surgery	4.16 (3.4)	5.42 (3.9)	3.75 (3.1)	*t =* −2.98; *p =* 0.009
Marital status				
Single	45.4% (44)	70.6% (12)	40% (32)	X^2^ = 7.90; *p* = 0.078
Married/Civil union	48.4% (47)	23.5% (4)	53.7% (43)	
Divorced	4.1% (4)	-	5% (4)	
Widowed	2.1% (2)	5.9% (1)	1.3% (1)	
Education level				
Basic education	2.1% (2)	-	2.5% (2)	X^2^ = 5.83; *p* = 0.44
Mid-level education	26.8 (26)	29.4% (5)	26.3% (21)	
Higher education	71.1% (69)	70.6% (12)	71.2% (57)	
Type of surgery				
Gastric bypass	51.5% (50)	41.2% (7)	53.8% (43)	X^2^ = 0.887; *p* = 0.346
Sleeve gastrectomy	48.5 (47)	58.8% (10)	46.2% (37)	

**Table 4 ijerph-19-07107-t004:** Means (M), standard deviation (SD), and results of the one-way multivariate analysis, for self-efficacy and LC, in accordance with current weight status.

Variables	Maintaining M (SD)	Regaining M (SD)	F(p)	η^2^
Age	40.6 (10)	35.53 (10.63)	3.64 (0.59)	0.037
Sex	91.3% (73)	82.4% (14)	1.19 (0.278)	0.012
Years since surgery	3.75 (3.1)	5.42 (3.9)	4.37 (0.039)	0.044
Self-efficacy for eating	69.1 (15.2)	57.70 (17.04)	7.48 (0.007)	0.073
Self-efficacy for exercising	49.1 (18.4)	37.82 (16.6)	5.39 (0.022)	0.054
Locus of control for eating	48.2 (6.6)	40.5 (8.6)	17.51 (0.001)	0.156
Locus of control for losing weight	43.3 (5.4)	36.4 (7.4)	20.12 (0.001)	0.175

**Table 5 ijerph-19-07107-t005:** Multiple logistic regression analysis.

Variables	Odds Ratio	95% CI	*p* Value
Age	0.860	0.779–0.950	0.003
Sex	1.665	0.238–11.64	0.608
Years since surgery	1.404	1.117–1.766	0.004
Self-efficacy for eating	0.991	0.932–1.054	0.778
Self-efficacy for exercising	0.993	0.943–1.045	0.775
Locus of control for eating	1.072	0.902–1.274	0.429
Locus of control for losing weight	0.760	0.606–0.953	0.018

**Table 6 ijerph-19-07107-t006:** Topics and subtopics analyzed.

Topic	Subtopic
Weight history prior to the surgery	- Explanations for being overweight
	- Explanations regarding the outcomes of other weight-loss methods
Motivation/expectations regarding the bariatric process	- Motivation/expectations at the start of the bariatric process
	- Evolution of expectations associated with the surgery
Current status	- Explanations regarding the current situation
	- Stages of change

**Table 7 ijerph-19-07107-t007:** Characterization of participants in the semi-structured interviews.

Code	Sex	Month and Year of the Surgery	Age at Surgery	Current Age	Type of Surgery	Height (cm)	Previous Weight	Weight at Nadir	Current Weight
1NNR	F	March 2016	34	36	Gastric bypass	1.58	148	77	83
2NRE	F	June 2016	63	65	Gastric bypass	1.61	126	70	96
3NNR	F	March 2011	23	30	Gastric bypass	1.55	101	59	73
4NNR	M	August 2006	44	56	Gastric bypass	1.82	190	85	95
5NNR	F	August 2016	46	48	Gastric bypass	1.70	123	75	79
6NNR	F	August 2015	42	45	Sleeve gastrectomy	1.63	95	55	60
7NRE	M	December 2002	24	39	Gastric bypass	1.64	136	74	115
8NRE	F	July 2005	41	54	Sleeve gastrectomy	1.60	116	78	113
9NNR	F	October 2015	35	38	Sleeve gastrectomy	1.64	118	74	77
10NNR	F	March 2016	52	54	Gastric bypass	1.50	89	47	50
11NNR	F	April 2017	48	49	Gastric bypass	1.54	105	55	58
12NNR	F	November 2016	38	40	Sleeve gastrectomy	1.64	104	60	62
13NRE	F	August 2014	34	38	Sleeve gastrectomy	1.62	98	56	88

Note: in the codes, NNR% indicates maintenance of the TWL%, and NRE indicates regaining of 15% or more of the TWL%.

## Data Availability

Not applicable.

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
