# Peer review of "Relationship among Self-Efficacy Expectations, Locus of Control, and Attributions in Bariatric Patients"

_ijerph, 2022, doi:10.3390/ijerph19127107_

Round 1

Reviewer 1 Report

Dear Authors, 

the manuscript is overall well written. The introduction nicely informs about the relevant background. Here are some suggestions: 

  • Please describe your findings in a more concrete manner in the results section of he abstractn(maybe with some numbers)
  • How were subjects selected for the qualitative analysis? Have everyone been asked?
  • Could you please further justify your sample size approach.
  • Is there literature supporting your definition of regaining / maintining. Were your overall results sensitive to this definition? (eg sensitive to defining a cut off of 10% or 20%)
  • Please provide totals in Table 3
  • Why were there so many females in both groups? Is this related to the study design or the true patient population?
  • Table 3: What is “BMI- previous” exactly referring to?
  • How long was the period between surgery and the follow-up? Where there group differences?
  • Table 5: You have chosen the "Correct Percentage" to describe the accuracy of your prediction model. However, this parameter is somewhat "unfair" because it depends strongly on the ratio of the two outcomes (e.g. always predicting "maintain" would also have had an accuracy of 82%). Parameters like sensitivity/specificity or PPV/NPV are more informative.

Author Response

  1. Abstract: the participants and results section of the summary has been modified.
  2. Participants: We have added information on qualitative sampling in the participants section.
  3. Definition of regaining / maintining: Added two references to the definition of re-gain and weight maintenance
  4. Table 3: Added totals
  5. % Females: The characterization of the sample is mostly female (89.7%). This is due to the fact that the population of patients who voluntarily answered the questionnaire had this sociodemographic characteristic. This reflects the true patient population.
  6. BMI- previous: was modified by pre-surgical BMI.
  7. How long was the period between surgery and the follow-up? Where there group differences? This information has been added to Table 3. We have also added to the analyses the sociodemographic variables evaluated and the postoperative follow-up time.
  8. Correct percentage: modified by parameters such as sensitivity/specificity or PPV/PVR/VNP.

Reviewer 2 Report

Thank you for the opportunity to review this paper. I found the study interesting and I always appreciate a mixed methods exploration of complex processes. Overall, I think this is a good study that makes a nice contribution to the literature, but has some methodological and conceptual flaws that should be addressed (or at least explained) prior to publication.

line 30: I’m not sure the word “fail” here is appropriate? it also carries some negative connotations that are not helpful to the topic

line 34: these can likely be combined into a single sentence: …explaining weight regain, defined as….”

line 37: I’m not familiar with the phrase “cognitive-attitudinal nature”, is there a better way to express this concept? I did a google search for exact match and only found a single hit, in a paper from 1982.

line 36-40: this paragraph is a single, very long run on sentence.

Line 103: How was this data collected? Did you ask patients, extract from medical records, both?

Methods: I don’t fully understand from the text why a top down coding approach was implemented? I would have thought that an open coding, phenomenology approach might have allowed the investigators to explore other possible theories or explanations? I’m less familiar with top down; perhaps the authors could explain this choice.

Kappa values are reported; I assume this is for first round comparison and then the independent reviewers discussed each area of disagreement until consensus was reached? otherwise, without a consensus process, how do you know what the final code is?

Table 3: when is “BMI - previous” is this immediately pre-operative BMI?

line 262: the word “mainly” doesn’t make sense to me here - the word mainly typically implies 1 or 2 outcomes, so to say “mainly as a means of achieving diverse results” doesn’t really make sense.

Discussion:

The concept of locus of control is interesting, nad exploring how patients might have internal vs external locus of control. I think the discussion would benefit from speaking to what we know about social determinants of health, genetics, and pathophysiologic derangement of central emergency balance homeostasis, and how for some patients, they may genuinely not have that much control over their weight through behavior change alone. discussing behavior in a vacuum is counter productive nad ignores the biological components of obesity.

Author Response

  1. Fail: Indeed, “fail” is not an appropriate word. It was modified.
  2. Line 34: was combined into a single sentence
  3. Line 37: The "cognitive-attitudinal nature" was modified by psychological variables related to behavioral self-regulation.
  4. Line 36-40: the sentence was modified
  5. Line 103 – now line 110: information on how the data were obtained was added
  6. Methods: we add a justification for the selection Top-down coding
  7. Kappa values: Information on how the codes have been chosen has been added.
  8. BMI- previous: was modified by pre-surgical BMI.
  9. line 262 /now 275: the word "mainly" was deleted.
  10. Discussion: we add a paragraph at the beginning of the discussion alerting the reader to the limitation of a non-multidimensional analysis of weight and obesity.

Kind regards

Round 2

Reviewer 1 Report

Thanks for the revision. 

Author Response

Dear reviewer,

We hope you are well.

We have included in the revision of our article the detailed aspects according to your suggestions. If there is something more specific you require, please let us know.

  1. Abstract: the participants and results section of the summary has been modified.
  2. Participants: We have added information on qualitative sampling in the participants section.
  3. Definition of regaining / maintining: Added two references to the definition of re-gain and weight maintenance
  4. Table 3: Added totals
  5. % Females: The characterization of the sample is mostly female (89.7%). This is due to the fact that the population of patients who voluntarily answered the questionnaire had this sociodemographic characteristic. This reflects the true patient population.
  6. BMI- previous: was modified by pre-surgical BMI.
  7. How long was the period between surgery and the follow-up? Where there group differences? This information has been added to Table 3. We have also added to the analyses the sociodemographic variables evaluated and the postoperative follow-up time.
  8. Correct percentage: modified by parameters such as sensitivity/specificity or PPV/PVR/VNP.

Kind regards,

Reviewer 2 Report

Thank you for your responses to my previous suggestions. In general, I think this revision of the paper reads better, and all my comments have been addressed satisfactorily, except where noted below. a few additional questions/comments:

1. Line 100: "Thirteen people agreed to be interviewed and it was considered that 100 a percentage close to 20% of the total sample was sufficient for this analysis" Where does this concept come from? in all types of qualitative research I'm familiar with, thematic saturation is what determines a sample size, not some pre-determined proportion. Could you please justify this?

2.  I still dont think that the authors have justified why they used a deductive (top-down) coding approach in the first place. For this type of research I would expect either an inductive approach, or a hybrid approach. the current edits provide more context, but still not a justification. I am not advocating that the authors go back and reanalyze their data. Rather I think that the others need to be able to convincingly explain why they chose a deductive coding approach which is known to be limited. I also think they need to include this as a limitation (the inherent limitations of deductive vs. inductive coding)

Author Response

 Dear Reviewer

We hope you are well.

We have added information as requested.

Furthermore, with regard to your comment on the limitation, we have not added it, as we consider that having based the coding on three well-established and well-founded models is a strong argument for using top-down coding.

What has been included is detailed below:

  1. Line 100. We add: as the results of the qualitative analysis allowed us to identify and analyse the explanations patients have for their current state of weight maintenance or weight regain.
  2. Justified why they used a deductive (top-down)

A justification was added based on the objective of the research.

We add:

On this basis, the Top-Down coding method was considered the most appropriate to identify and analyse whether patients' narrative data could be understood within the framework of Weiner's attribution theory [31], Bandura's self-efficacy theory [16] and stages of change [44]. 

Kind regards